# Modelling the impact of lockdown-easing measures on cumulative COVID-19 cases and deaths in England

Hisham Ziauddeen,[1,2,3] Naresh Subramaniam,[1] Deepti Gurdasani  [4]

¹Dept. of Psychiatry, University of Cambridge, Cambridge, UK
²Wellcome-MRC Institute of Metabolic Science, University of Cambridge, Cambridge, UK
³Cambridgeshire & Peterborough Foundation Trust, Cambridge, UK
⁴William Harvey Research Institute, Queen Mary University of London, London, UK

**Correspondence to**
Dr Deepti Gurdasani;
d.gurdasani@qmul.ac.uk

## ABSTRACT

**Objectives** To assess the potential impacts of successive lockdown-easing measures in England, at a point in the COVID-19 pandemic when community transmission levels were relatively high.

**Design** We developed a Bayesian model to infer incident cases and reproduction number ($R$) in England, from incident death data. We then used this to forecast excess cases and deaths in multiple plausible scenarios in which $R$ increases at one or more time points.

**Setting** England.

**Participants** Publicly available national incident death data for COVID-19 were examined.

**Primary outcome** Excess cumulative cases and deaths forecast at 90 days, in simulated scenarios of plausible increases in $R$ after successive easing of lockdown in England, compared with a baseline scenario where $R$ remained constant.

**Results** Our model inferred an $R$ of 0.75 on 13 May when England first started easing lockdown. In the most conservative scenario modelled where $R$ increased to 0.80 as lockdown was eased further on 1 June and then remained constant, the model predicted an excess 257 (95% CI 108 to 492) deaths and 26 447 (95% CI 11 105 to 50 549) cumulative cases over 90 days. In the scenario with maximal increases in $R$ (but staying ≤1), the model predicts 3174 (95% CI 1334 to 6060) excess cumulative deaths and 421 310 (95% CI 177 012 to 804 811) cases. Observed data from the forecasting period aligned most closely to the scenario in which $R$ increased to 0.85 on 1 June, and 0.9 on 4 July.

**Conclusions** When levels of transmission are high, even small changes in $R$ with easing of lockdown can have significant impacts on expected cases and deaths, even if $R$ remains ≤1. This will have a major impact on population health, tracing systems and healthcare services in England. Following an elimination strategy rather than one of maintenance of $R$ ≤1 would substantially mitigate the impact of the COVID-19 epidemic within England.

## Strengths and limitations of this study

► This study provides urgently needed information about the potential impact of successive lockdown-easing measures in England when community transmission of SARS-CoV-2 is relatively high.

► We use a robust Bayesian model based on Office of National Statistics-registered deaths in England, to infer incident cases and reproduction number and then forecast deaths and cases considering multiple plausible scenarios of increase in reproduction number with successive easing of lockdown in England.

► Our study focuses on the impact of easing lockdown in the conservative scenario when $R$ is maintained at or below 1 in line with stated government policy, showing that even this scenario would result in substantial excess of cases and deaths relative to a baseline scenario of not easing lockdown or elimination.

► The excess cumulative cases are likely to be sensitive to the specified infection fatality ratio, although this is not expected to materially change the results and inferences. We have assumed a constant infection fatality rate across time, which would not account for changes in the age composition of the infected cases over time.

► The model inference is dependent on reliable reported statistics on incident deaths. Underestimation of recent registered deaths would lead to more conservative $R$ inference, and underestimation of the impact of easing lockdown.

## INTRODUCTION

As countries around the world negotiated the first wave of the COVID-19 pandemic, governments had to make critical decisions about when and how they eased the lockdown measures instituted to control the pandemic. Given the significant risks of a resurgence of the pandemic and the consequent implications, these decisions have had important consequences on pandemic control following easing of lockdown restrictions globally.

Different countries eased lockdown in different ways and at different points in their epidemic trajectory.[1] The UK imposed lockdown relatively late in its epidemic trajectory and began easing lockdown relatively early, when community transmission levels (incident cases) were still high.[2] By contrast, Germany, Denmark, Italy and Spain started easing lockdown when incident cases and deaths were at much lower levels. However, despite mitigation strategies such as test, trace

and isolation systems in place, countries like Germany saw increases in reproduction number (*R*) after easing lockdown, with increases to above 1 in June.[3] South Korea and China too saw a resurgence in new cases after easing their lockdowns and went on to put in place localised restrictions to control the spread of infections.

Several experts, including the UK government's Scientific Advisory Group for Epidemics (SAGE), cautioned against easing lockdown in May 2020,[2] when community transmission was still high, warning that this could overwhelm the still nascent testing and contact tracing services that could mitigate the impact of easing lockdown and greatly impact the health service. Nevertheless, the UK proceeded with easing lockdown with the stated aim of doing so while keeping *R* ≤1. On 13 May, people who could not work from home were asked to return to work. On 1 June, schools were reopened, outdoor markets and showrooms opened and households were allowed to meet in socially distanced groups of six. On 15 June, non-essential businesses, including the retail sector, were opened. In the week of 29 June, a surge in cases was reported in Leicester, England, leading to the reimposition of restrictive measures and concern that other regions in England may experience similar increases in case numbers.[4] Nevertheless, the government went ahead with the next planned easing of lockdown on 4 July, when pubs, cafes and hotels opened.

As the country proceeded to rapidly ease lockdown, it was vital to understand and quantify the potential impact of this so as best inform public health strategy. In June 2020, we modelled these impacts across a range of plausible scenarios over the 90-day period from 1 June to 29 August. Using an epidemiological model of COVID-19 spread with Bayesian inference, we inferred parameters of the epidemic in England using daily death data from the Office of National Statistics (ONS). We estimated the time-varying *R* and daily cases, and then used these to forecast cases and deaths in several plausible scenarios in which *R* increased with the easing of lockdown, particularly focusing on those in which *R* remained ≤1, and contrasted these with elimination strategies that aim to suppress *R* as much as possible.

During the manuscript review process, we were able to examine the observed data that accrued through the original forecasting period and compare it against the model predictions.

## METHODS
The original model inference and forecasting were carried out in June 2020 and the model development is described below. Following this, we describe the comparison of the model predictions from the original forecasts with the observed data from the forecasting period.

### Data for model development
In order to model the impact of easing lockdown, we needed to know the levels of transmission and growth parameters of the regional epidemic. Given the limited community testing and case detection in the UK, incident case numbers at that point were likely to be substantially underestimated. We therefore based our model on the number of incident deaths by date of occurrence, which is likely to be more reliable.[5] Incident deaths are a function of incident cases in the previous weeks and the reproduction rate of the epidemic, and both these parameters can be inferred from the death data.[5] We included data until 12 June for England, as released by the ONS on 30 June 2020 (25th week of published data).[6] These data are based on deaths registered by 27 June. As reporting delays mean that more recent deaths are underestimated, we only considered deaths up to 12 June.

### Patient and public involvement
As only publicly available aggregate incident death statistics were used, there was no direct patient or public involvement.

## PRIMARY OUTCOMES
We assessed the excess cumulative predicted cases and deaths, over a 90-day period from 1 June. We assumed different scenarios of changing *R* at the points of lockdown easing, in comparison with a baseline scenario in which *R* remained constant during this period.

### Estimation of incident cases
Incident cases and time-varying *R* numbers were estimated using a Bayesian model, similar to that previously described by Flaxman *et al*,[5] accounting for the delay between onset of infection and death. The number of infected individuals is modelled using a discrete renewal process, as has been described before.[5] This is related to the commonly used Susceptible–Infected–Recovered model, but is not expressed in differential form.

We modelled cases from 30 days prior to the first day that 10 cumulative deaths were observed in England, similar to previous methods.[5] The numbers of incident cases for the first 6 days of this period were set as parameters to be estimated by the model (table 1).

Subsequent incident case numbers would then be a function of these initial cases and estimated *R* values. We assumed a serial interval (SI) with a lognormal distribution with mean 4.7 and SD of 2.9 days, as in Nishiura *et al*.[7] The SI was discretised as follows:

$$g_s = \int_{t=s-1}^{s} g(t)\, dt$$

For s=1,2…N, where N is the total number of intervals (each interval being 1 day) estimated. We estimated the distribution for 201 days, to align with the 111 days of data up to 29 May, plus 90 days of forecasting. Given an SI distribution, the number of infections $C_t$ on a given day *t* is given by the following discrete convolution function:

$$c_t = R_t \sum_{j=0}^{t-1} c_j g_{t-j}$$

**Table 1** Parameters estimated by Bayesian model

| Variable | Parameter | No | Priors |
|---|---|---|---|
| $c_t$, where t=1…6 | Number of initial cases on first 6 days | 6 | Exponential(1.0/tau) |
| $R_0$ | Baseline reproduction number | 1 | Normal(2.4,0.5) |
| $R_t$ | Time-varying effective reproduction number | 9 | Normal(0.8,0.25) |
| φ | Variance parameter for negative binomial distribution of deaths | 1 | Normal(0,5) |
| τ | Parameter in prior of $c_t$ | 1 | Exponential(0.03) |

The incident cases on a given day $t$ are therefore a function of $R$ at point $t$ and incident cases up to time $t–1$, weighted by the distribution of the SI.

### Estimation of time-varying reproduction number

The baseline reproduction number ($R_0$), and the subsequent time-varying effective reproduction number ($R_t$) were estimated up to 12 June. We allowed $R_t$ to change on at least three points: (1) 16 March, when the UK first introduced social distancing measures; (2) 23 March, when lockdown measures came into place with stay-at-home instructions and closures of schools and non-essential businesses; and (3) 13 May, the first easing of lockdown. We also considered models in which $R_t$ was allowed to change on 1 June. Given the limited death data, that is, only up to 12 June, we were unlikely to be able to estimate changes in $R_t$ after 13 May with sufficient certainty. Observed deaths from 1 June are likely to be a function of cases 2–3 weeks prior to this, and were unlikely to reflect changes in $R_t$ from 1 June.

### Model selection

We assessed and compared models that allowed $R_t$ to change at the four points described above (model 1), with more flexible models that allowed more frequent changes (models 2 and 3), as follows:

1. Model 1: 16 March, 23 March, 13 May and 1 June.
2. Model 2: every week from the beginning of the modelling period, including on 16 March, 23 March, 13 May and 1 June.
3. Model 3: 16 March, 23 March and 13 May, and every week between 23 March and 13 May, that is, during lockdown.

For each model, we used the R package *loo* to calculate expected log pointwise predictive density (ELPD) using leave-one-out (LOO) cross-validation individually for each left-out data point based on the model fit to the other data points. We then calculated between-model differences in ELPDs, to assess whether particular models predicted data better than others, as discussed previously.[8] As the assumptions in estimation of ELPD may be violated given these are time-series data, and therefore correlated, we also compared the root mean squared errors (RMSEs) across models to assess fit. The final model used was arrived upon based on these comparisons, prioritising differences in ELPD, as this has been used in a similar context to assess change points, previously.[9] We assessed

whether models were significantly different (ELPD difference/SE of difference >2). When models were not statistically significantly different in performance, for simplicity, we prioritised the model where the least number of parameters needed estimation.

In addition, we also compared model 1 (four change points) with models where each of the change points was left out in turn, as done by Dehning *et al*,[9] to assess if these dates do correspond to change points in $R_t$.

### Estimation of deaths

Incident deaths from COVID-19 are a function of the infection fatality rate (IFR), the proportion of infections that result in death and incident cases that have occurred over the past 2–3 weeks. For observed daily deaths ($D_t$) for days $t \in 1, …, n$, the expectation of observed daily deaths ($d_t$) is given by:

$$d_t = E(D_t)$$

As described in Flaxman *et al*,[5] we model the number of observed daily deaths $D_t$ as following a negative binomial distribution with mean $d_t$ and variance $d_t + \frac{d_t^2}{\psi}$, where ψ follows a half normal distribution:

$$D_t \sim Negative\ Binomial\left(d_t,\ d_t + \frac{d_t^2}{\psi}\right),$$

where $\psi \sim Normal^+(0,5)$.

Similar to estimation of incident cases, deaths at time point $t$ ($d_t$) were modelled as a function of incident cases up to time $t–1$, weighted by the distribution of time of infection to time of death (π). The π distribution was modelled as the sum of the distribution of infection onset to symptom onset (the incubation period), and the distribution of symptom onset to death. As has been previously done,[5] both of these were modelled as gamma distributions with means of 5.1 days (coefficient of variation 0.86) and 18.8 days (coefficient of variation 0.45), respectively, as follows:

$$\pi \sim IFR * \left(Gamma(5.1,\ 0.86) + Gamma(18.8,\ 0.45)\right)$$

IFR was assumed to be 1.1%, based on the most recent estimates from the University of Cambridge MRC nowcasting and forecasting model.[10] This estimate is in line with estimates from Flaxman *et al* (Imperial), of 1% that have been widely used in modelling of COVID-19 deaths across the UK.[5] These estimates are based on those reported by Verity *et al*,[11] during early epidemiological inference from the outbreak in Wuhan, and are corrected for age structure and contact patterns for the UK, as previously outlined.[5] Misspecification of the IFR estimate

would lead to biased inference of case numbers, but not deaths, as this can be considered as a scaling factor, that is, used first to estimate the cases, which are then used to accurately predict observed deaths and future deaths based on different scenarios. Therefore, the predicted death numbers can be thought of as independent of these estimates. For simplicity, we consider a fixed IFR over time.

To discretise the time to death distribution, we estimated the probability of death within each discrete time interval (1 day), conditional on surviving previous intervals. First, we calculate the hazard ($h_t$), the instantaneous probability of failure (ie, dying) within a time interval, as follows:

$$h_t = \frac{\int_{t=s-0.5}^{s+0.5} \pi\left(t\right) dt}{1 - \pi_{s-0.5}}$$

As the denominator excludes individuals who have died, this ensures that $h_t$ is calculated only among those surviving. The probability of survival within each interval is:

$$s_t = 1 - h_t$$

The cumulative survival probability of surviving up to the interval $t–1$ is therefore:

$$S_{T>t-1} = \prod_{j=1}^{t-1} s_j$$

where $T$ is the time of death of an individual. In other words, the cumulative probability of survival up to interval $t$ is simply the product of survival within each interval up to $t–1$, where the probability of survival within each interval ($s_t$) is $1–h_t$, where $h_t$ is the probability of dying within that interval.

Given this, we now estimate the probability of death within interval $t$, conditional on surviving up to $t–1$ as:

$$\omega_t = P\left(T = t \mid T > t - 1\right) = S_{T>t-1} * h_t$$

Here, $\omega$ represents the discretised distribution of infection onset to death, with the probability of death within interval $t$ conditional on surviving previous intervals. Deaths can therefore be calculated as a function of incident cases of infection within previous intervals, as follows:

$$d_t = \sum_{j=0}^{t-1} c_j \omega_{t-j}$$

Here, the number of deaths within interval $t$ (on a given day) is a sum of the number of daily cases up to the previous day, with previous cases weighted by the discretised probability distribution of time from onset of infection to death.

### Estimated parameters and model priors

We estimated the set of model parameters $\theta$={$c_{1-6}$, $R_0$, $R_t$, $\varphi$, $\tau$} using Bayesian inference with Markov chain Monte Carlo (MCMC) (table 1). We estimated the number of cases in the first 6 days of the modelled period, as

subsequent cases are simply a function of cases on these days, the SI and $R_t$. As described above, $R_0$ was constrained up to 16 March and then again after 13 May. For the period prior to 16 March, we assigned a normal prior for $R_0$ with mean 2.5 and SD 0.5. For the period that $R_t$ was allowed to vary, that is, every week from 16 March until 13 May, we assigned a normal prior with a mean 0.8 and SD 0.25. These priors are based on estimates of time-changing $R_t$ from the University of Cambridge MRC biostatistics nowcasting and forecasting models[10] and SAGE estimates of $R$,[12] and consistent with Flaxman *et al.*[5] For the number of cases on day 1, we assigned a prior exponential distribution:

$$y \sim exponential\left(\frac{1}{\tau}\right)$$

where $\tau \sim exponential\left(0.03\right)$

### Model estimation

Parameters were estimated using the Stan package in R with MCMC algorithms used to approximate a posterior distribution of parameters by randomly sampling the parameter space. We used 4 chains with 1000 warm up samples (which were discarded), and 3000 subsequent samples in each chain (12 000 samples in total) to approximate a posterior distribution using the Gibbs sampling algorithm. From these, we obtained the best-fit values and the 95% credible intervals (CIs) for all parameters. We used these parameters to estimate the number of incident cases and deaths in England. We examined the fit-of-the-model predicted deaths to the observed daily deaths from the ONS, and also the consistency of the model parameters with known values in the literature, estimated from global data. We assessed the distribution of *R-hat* values for all parameters, to assess convergence between chains.

### Sensitivity analyses

We carried out sensitivity analyses using uninformative priors for $R_0$ and $R_t$, to examine the sensitivity of $R_t$ estimates to prior specification. We also examined the impact of the SI by comparing the baseline model (SI of mean 4.7 and SD 2.9 days) with a longer SI modelled as a gamma distribution with mean 6.5 and coefficient of variation of 0.72, as estimated by Chan *et al.*[13]

### Forecasting cases and deaths

All forecasts were carried out up to 90 days (29 August 2020) after 1 June. We considered a set of scenarios in which $R_t$ increased from baseline on 1 June and then remained constant, as well as those in which further increases in $R_t$ occur on 15 June and 4 July. We considered an increase in $R_t$ of up to 0.25 in increments of 0.05, this being a plausible degree of change in response to easing lockdown, based on the empirical data from other countries,[3 14] as well as the modelling by UK SAGE.[15] Finally, for comparison with a strategy of elimination, namely suppressing $R_t$ to the lowest level possible before easing lockdown measures, as has been done in South Korea,

New Zealand and Australia, we also modelled scenarios with $R_t$ values of 0.6 and 0.7.

For each of these scenarios, we predicted the number of incident cases and incident deaths, using the functions from the inference model above. Briefly, cases are a function of $R_t$, incident cases on previous days and the SI discretised distribution:

$$c_t = R_t \sum_{j=0}^{t-1} c_j g_{t-j}$$

Deaths are a function of incident cases over previous weeks, and the distribution of onset of infection to death times:

$$d_t = \sum_{j=0}^{t-1} c_j \omega_{t-j}$$

All scenarios were compared with a baseline scenario of no change in $R_t$ from 13 May onwards.

### Comparison of model predictions with observed data

The observed death data for daily deaths in England up to 28 August as obtained from the ONS (from data up to 11 September) were plotted against the original model predictions from June, and the RMSE was calculated between the observed data and the predicted deaths in the different modelled scenarios. The model was rerun with these data, to infer values of $R_t$ until 28 August. As the purpose of this exploratory model was inference of parameters, $R_t$ was allowed to change weekly from 16 March, as well as at time points of easing lockdown: 13 May, 1 June, 15 June and 4 July as in the original forecasting and 25 July (gyms and pools reopened), and 15 August (casinos, bowling alleys and soft play areas reopened). Where these dates fell on the weekly change point, they were not included separately.

### RESULTS
### Model selection and model inferences

Model 3, which allowed weekly changes in $R_t$ during lockdown, produced the best fit to the data (online supplemental table 1), with estimation of fewer parameters compared with model 2. This was therefore used as the primary model and unless otherwise stated, all inferences described subsequently are from this model.

We inferred $R_0$ of 3.65 (95% CI 3.36 to 3.96), consistent with previous estimates within the UK.[5] The $R_t$ is estimated to have declined substantially following initiation of social distancing and lockdown measures, reaching a low of 0.66 (95% CI 0.34 to 1.04) during the week 30 March–5 April 2020. The most recent $R_t$ from 13 May is estimated as 0.752 (95% CI 0.50 to 1.00) (figure 1). The alternative models allowing change of $R_t$ on 1 June inferred a very similar $R_t$ for 1–12 June suggesting that there were insufficient data to accurately infer any changes to $R_t$ following the easing of lockdown on 1 June. On examining the impact of constraining $R_t$ on model fit at any of the four change points, this appears greatest for 16 March (when social

**Figure 1** Estimated time-varying reproduction number ($R_t$) for England. The figure shows the $R_t$ estimated by model 3 (blue) with 95% credible intervals (CIs) (grey) with a serial interval of mean 4.7 and SD 2.9 days. From 3.65 (95% CI 3.36 to 3.96), $R_t$ drops on 16 March and 23 March (indicated by vertical dashed lines) when social distancing and lockdown were instituted, reaching a low of 0.66 (95% CI 0.34 to 1.04) in the week of 30 March. The last estimated $R_t$ is 0.75 (95% CI 0.50 to 1.00) following 13 May.

distancing measures were put into place) (online supplemental table 2) with only modest impacts on model fit of constraining $R_t$ on 23 March and 13 May, and no impact on constraining $R_t$ on 1 June.

The model showed a good fit to the observed distribution of deaths up to 12 June (figure 2). R-hat estimates were <1.05 for all estimated parameters (online supplemental figure 1). LOO cross-validation also supported a good model fit, with the shape parameter k <0.5 for all values (online supplemental figure 2). The median number of incident cases inferred on 1 June was 4317/day (95% CI 2062 to 8155), which was broadly consistent with the estimates from the ONS survey for England based on a random sample of the population within the same time period.

### Forecasts of lockdown-easing scenarios

In the baseline forecasting scenario where $R_t$ remained constant ($R_{test}$=0.75) through the 90-day forecasting period (1 June–29 August 2020), the model predicted 48 501 (46 170–50 989) cumulative deaths in England (online supplemental table 4). By comparison, the ONS reported 46 539 cumulative deaths up to 12 June in England (registered up to 27 June).

In the scenarios where $R_t$ increased on 1 June and then remained constant, for increases from the median 0.75–0.80, 0.85, 0.90, 0.95 and 1, the model predicted median

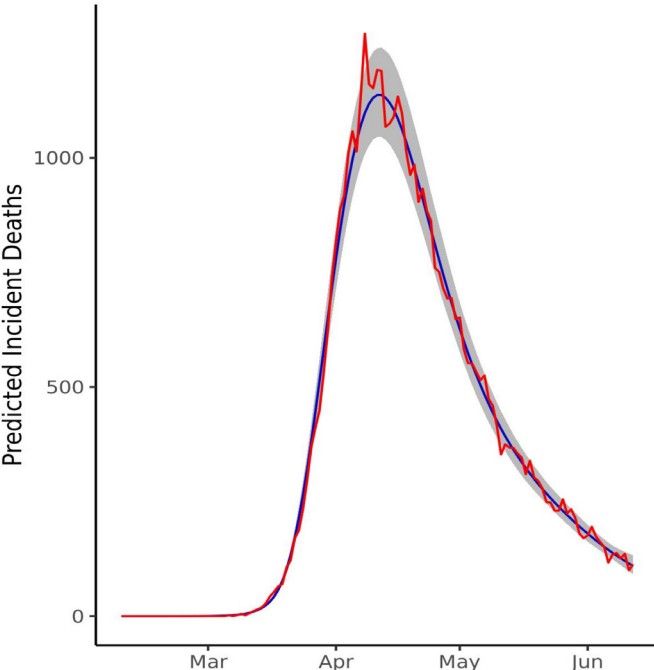

**Figure 2** Model fit to observed death data. Daily deaths predicted by model 3 (blue) with 95% credible intervals (grey) show a good fit to the observed deaths from the ONS (red). ONS, Office of National Statistics.

excess deaths of 257 (95% CI 108 to 492), 632 (95% CI 265 to 1208), 1173 (95% CI 493 to 2240), 1971 (95% CI 828 to 3764) and 3174 (95% CI 1334 to 6060), respectively. Increases of $R_t$ to 1.05 and 1.1, with resultant exponential growth, led to excess median deaths of 5017 (95% CI 2109 to 9578) and 7878 (95% CI 3313 to 15 037), respectively (figure 3 and online supplemental table 4).

In scenarios where $R_t$ increased on 1 June, 15 June and 4 July, we found that compared with the baseline scenario, modest increases of $R_t$ to 0.80, 0.85 and 0.90 on these dates, respectively, would lead to 508 (95% CI 213 to 972) excess deaths. If $R_t$ increased to 0.90, 0.95

and 1 at these time points, then excess estimated deaths increase to 1848 (95% CI 776 to 3534). In these scenarios, $R_t$ remains ≤1 (figures 3–5 and online supplemental table 4). Increases of $R_t$ above 1 at any point resulted in rapid increases in cases and deaths, with between 3600 and 13 000 excess deaths in different scenarios for $R_t$ rising up to between 1 and 1.2, predicting a second wave of the epidemic within England (figures 4 and 5 and online supplemental table 4).

Even in the conservative scenario where $R_t$ increased from 0.75 to 0.80 on 1 June and then remained constant thereafter, the model predicted an excess of 26 447 (95% CI 11 105 to 50 549) cumulative cases over 90 days. On the other hand, the scenario with the largest changes in $R_t$, but still remaining ≤1, predicted an excess of up to 421 310 (95% CI 177 012 to 804 811) (figures 6–8 and online supplemental table 4). For scenarios where $R_t$ rose beyond 1 (up to 1.2), we would expect between 540 000 and 2.8 million excess cases, in line with a second wave (online supplemental table 4).

### Forecasts from an elimination scenario

Compared with the baseline scenario of $R_t$ staying at 0.75, we found that maintaining $R_t$ at 0.60 and 0.70 would result in 44 302 (95% CI -84 684 to -18 600) and 19 968 (95% CI -38 168 to -8384) fewer cumulative cases, and 462 (95% CI -884 to -194) and 204 (95% CI -389 to -86) fewer deaths over the modelled 90-day period, respectively (figures 3 and 6, online supplemental table 4).

### Robustness of model in sensitivity analyses

Using uninformative (no prior specified) priors for $R_t$ did not materially alter the median estimates of $R_t$, although uncertainty around estimates was predictably increased (online supplemental figure 3). This suggests our estimates are robust to the priors specified.

Using a longer SI leads to an increase in the estimated $R_0$, although subsequent estimates following easing of

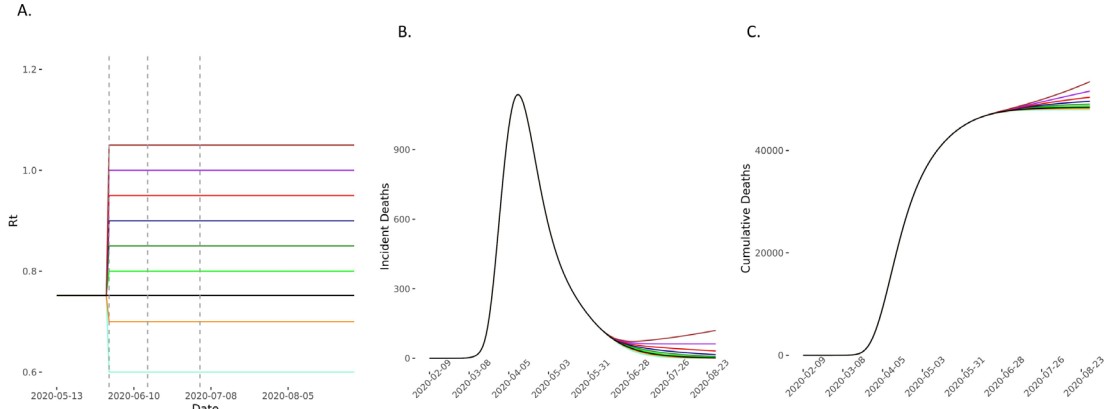

**Figure 3** Predicted deaths with $R_t$ increasing on 1 June. (A) The model compared scenarios in which $R_t$ increases to 0.80 (light green), 0.85 (green), 0.90 (dark blue), 0.95 (red), 1 (purple) and 1.05 (brown) and then remains constant for the 90-day forecasting period. The comparator baseline scenario is of $R_t$ remaining at 0.75 (black) and two elimination strategies of $R_t$ reducing to 0.7 (yellow) and 0.6 (light blue) were also considered. Vertical dashed lines represent time points of easing lockdown. (B,C) The incident and cumulative deaths increase in all scenarios in which $R_t$ increases and reduces in the two elimination scenarios. $R_t$, time-varying reproduction number.

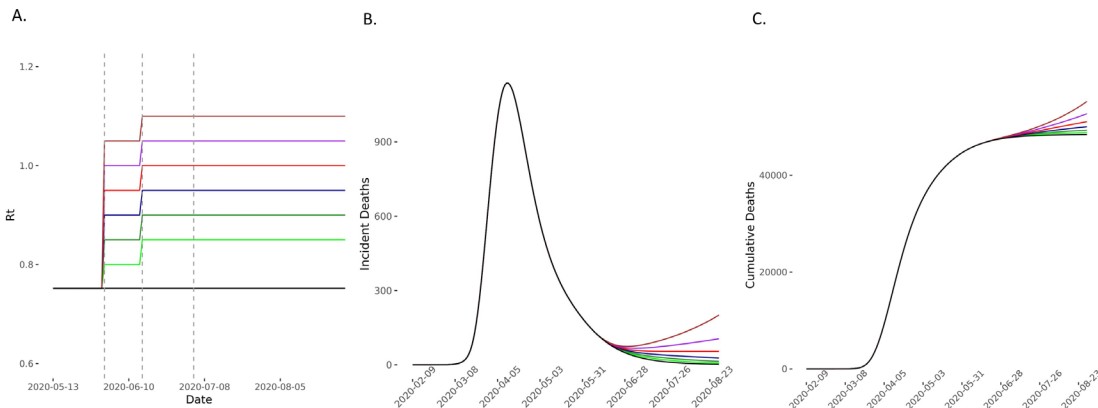

**Figure 4** Predicted deaths in scenarios of $R_t$ increase on 1 and 15 June compared with baseline scenario. (A) The model compared scenarios in which $R_t$ increases to 0.80 (light green), 0.85 (green), 0.90 (blue), 0.95 (red), 1 (purple) and 1.05 (brown) and then further by 0.05 on 15 June and then remaining constant for the 90-day forecasting period. The comparator baseline scenario is of $R_t$ remaining at 0.75 (black). Vertical dashed lines represent time points of easing lockdown. (B,C) The incident and cumulative deaths increase in all scenarios in which $R_t$ increases. $R_t$, time-varying reproduction number.

lockdown remain broadly comparable (online supplemental figure 4). This model is comparable with the primary model with regard to fit to observed deaths (online supplemental figure 5), although we note that predicted excess deaths and cases in all scenarios where $R_t$ <1.1 are higher than in the primary model with shorter SI (online supplemental table 4), suggesting the primary model is likely to be conservative.

### Comparison of model predictions with observed data

The observed cases and deaths are plotted against the modelled scenarios in figure 9. Among the scenarios studied, the observed daily deaths seems to align most closely with the scenario in which R values are 0.85, 0.85 and 0.9 at the three change points. The RMSE between the observed and predicted deaths is lowest for this scenario (online supplemental table 3). The inferred $R_t$ values concur with this (although uncertainty estimates are wide), and also suggest that it is in late July that $R_t$ started to creep above 1 (figure 10). We also note that

the observed cumulative deaths by 28 August represent an excess of 1291 deaths over our baseline scenario.

### DISCUSSION

In this paper, we describe a Bayesian model for inferring incident cases and reproduction numbers from daily death data, and for forecasting the impact of future changes in $R$. Our findings provide important quantification of the likely impact of relaxing lockdown measures in England, and to our knowledge, this is the first study to have comprehensively assessed this through several plausible scenarios. We show that even in scenarios in which $R$ remains ≤1 (in line with the UK government's stated aim), small increases in $R_t$ from lifting lockdown measures can lead to a substantial excess of deaths with 3174 (95% CI 1334 to 6060) in the most severe scenario modelled.

Our model inferences are robust to modelling assumptions of specified priors for $R_t$. We note, however, that

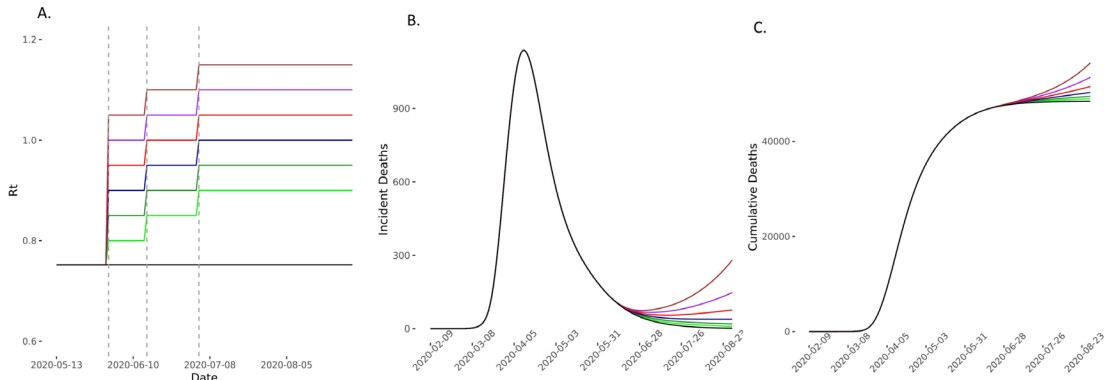

**Figure 5** Predicted deaths in scenarios of $R_t$ increase on 1 June, 15 June and 4 July compared with baseline scenario. (A) The model compared scenarios in which $R_t$ increases to 0.80 (light green), 0.85 (green), 0.90 (blue), 0.95 (red), 1 (purple) and 1.05 (brown) and then further by 0.05 on 15 June and then again by 0.05 on 3 July before remaining constant for the 90-day forecasting period. The comparator baseline scenario is of $R_t$ remaining at 0.752 (black). Vertical dashed lines represent time points of easing lockdown. (B,C) The incident and cumulative deaths increase in all scenarios in which $R_t$ increases. $R_t$, time-varying reproduction number.

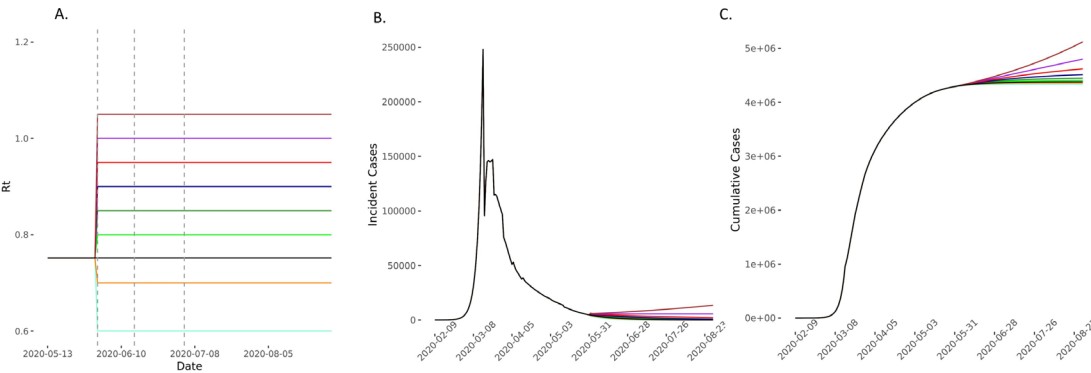

**Figure 6** Predicted cases in scenarios of $R_t$ increase on 1 June compared with baseline and elimination scenarios. (A) The model compared scenarios in which $R_t$ increases to 0.80 (light green), 0.85 (green), 0.90 (dark blue), 0.95 (red), 1 (purple) and 1.05 (brown) and then remains constant for the 90-day forecasting period. The comparator baseline scenario is of $R_t$ remaining at 0.752 (black) and two elimination strategies of $R_t$ reducing to 0.7 (yellow) and 0.6 (light blue) were also considered. Vertical dashed lines represent time points of easing lockdown. (B,C) The incident and cumulative cases increase in all scenarios in which $R_t$ increases and reduces in the two elimination scenarios. $R_t$, time-varying reproduction number.

using a longer SI would result in higher numbers of excess deaths for each scenario, suggesting that our primary scenario is conservative (online supplemental tables 3 and 4). Our estimated $R_t$ of 0.75 following 13 May is consistent with estimates from the SAGE group advising government at the time.[12] We assessed increases in $R_t$ that were entirely plausible, given the data from other European countries that have started easing lockdown.[3] Our model predicted a substantial excess of cases and deaths in several scenarios where $R$ remained ≤1, as well as scenarios where R increased up to 1.2. When we compared our predictions with the observed data from the original forecasting period, we found that these aligned most closely to the scenario in which R increased to 0.85 on 1 June, and then to 0.9 on 4 July. In contrast, our model showed that had an elimination strategy been pursued and $R_t$ suppressed to 0.6 or 0.7, this could have prevented a median estimated 462 and 204 deaths, and 44 302 and 19 968 cases, respectively, from the baseline scenario.

Countries like Denmark and Germany started easing lockdown when community transmission was low and this likely mitigated increases in $R$ with the lifting of lockdowns, alongside the use of aggressive case detection and contact tracing approaches. The UK began to ease lockdown when community transmission was still high (with daily estimated >8000 cases and >300 deaths) and still does not have a fully operational test, trace and isolate system at the time of writing, with the existing system overwhelmed by incident cases. The UK's current estimates of $R_t$ still rely on incident deaths (as used by the MRC nowcasting and forecasting model, and SAGE),[10] and therefore reflect community transmission from a median of 2–3 weeks ago.[12] Easing lockdown in two weekly steps meant that by the time we detected the impact of one step, the next one had already been instituted and not unexpectedly mitigating these impacts was challenging. At the time lockdown was being rapidly eased, UK SAGE expressed concerns that increases in $R$ up to 1.2 could continue undetected for longer periods of time.[15]

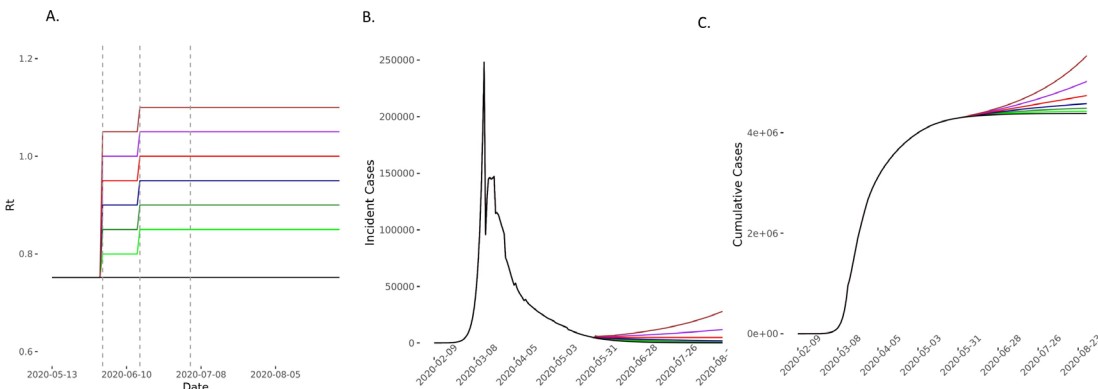

**Figure 7** Predicted cases in scenarios of $R_t$ increase on 1 June and 15 June compared with the baseline scenario. (A) The model compared scenarios in which $R_t$ increases to 0.80 (light green), 0.85 (green), 0.90 (blue), 0.95 (red), 1 (purple) and 1.05 (brown) and then further by 0.05 on 15 June and then remaining constant for the 90-day forecasting period. The comparator baseline scenario is of $R_t$ remaining at 0.752 (black). Vertical dashed lines represent time points of easing lockdown. (B,C) The incident and cases increase in all scenarios in which $R_t$ increases. $R_t$, time-varying reproduction number.

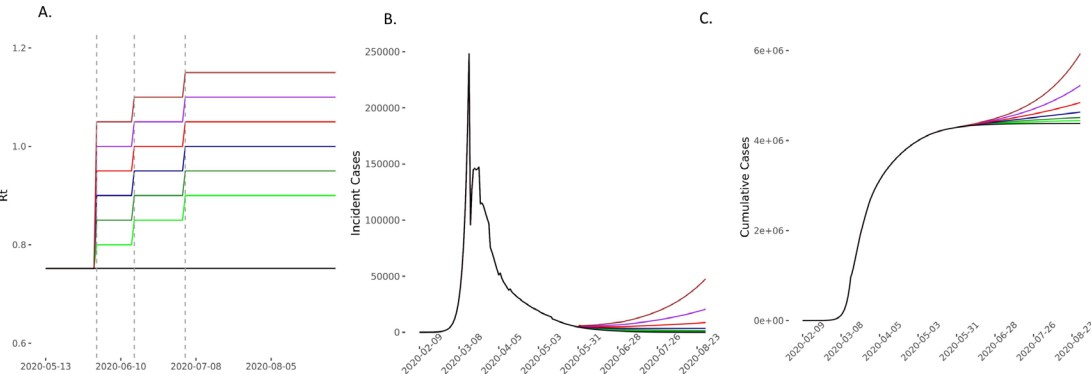

**Figure 8** Predicted cases in scenarios of $R_t$ increase on 1 June and 15 June and 4 July compared with the baseline scenario. (A) The model compared scenarios in which $R_t$ increases to 0.80 (light green), 0.85 (green), 0.90 (blue), 0.95 (red), 1 (purple) and 1.05 (brown) and then further by 0.05 on 15 June and then again by 0.05 on 3 July before remaining constant for the 90-day forecasting period. The comparator baseline scenario is of $R_t$ remaining at 0.752 (black). Vertical dashed lines represent time points of easing lockdown. (B,C) The incident and cumulative cases increase in all scenarios in which $R_t$ increases. $R_t$, time-varying reproduction number.

In September 2020, the UK is at point where community transmission is once again high and it is clear that we have entered the second wave of the pandemic. Schools reopened in the second week of September, a move that is vitally important to children's health and development, but one that can potentially increase community transmission. Cases and hospitalisations have been increasing exponentially, which has recently translated into an increase in weekly deaths. Using the best available confirmed COVID-19 case data in England published by the UK government on 21 September (which is likely an underestimate), we modelled the potential impact of increases in transmission on daily cases and deaths over the next 2 months, assessing different scenarios of increase in $R_t$. As $R_t$ reaches 1.5, the daily deaths approach 1000 by late November (figure 11). We note that the number of deaths forecast during this period could be overestimated if transmission is disproportionately higher among younger age groups, as overall IFR would be lower than the assumed 1%. However, as current data suggest,

transmission is likely to spill over into more vulnerable and older age groups over time. This has profound implications for the health service and the limited ICU (Intensive Care Unit) capacity available in the NHS (National Health Service), which is at great risk of being overwhelmed. Our modelling suggests that small changes in $R_t$ moving forward could have substantially large effects on case numbers and deaths, suggesting that mitigatory strategies implemented in a timely manner could have a large impact.

We acknowledge some important limitations of our model. The first is that it is based on a back calculation of cases based on incident deaths, which are likely to be underestimated due to reporting delays and under-reporting. Second, our model is reliant on inferring cases and reproduction numbers, which depend on the assumed distributions of the SI and the time of onset to death distributions. Though we based our assumptions on the literature, misspecification of these would influence our estimates. While we have evaluated this,

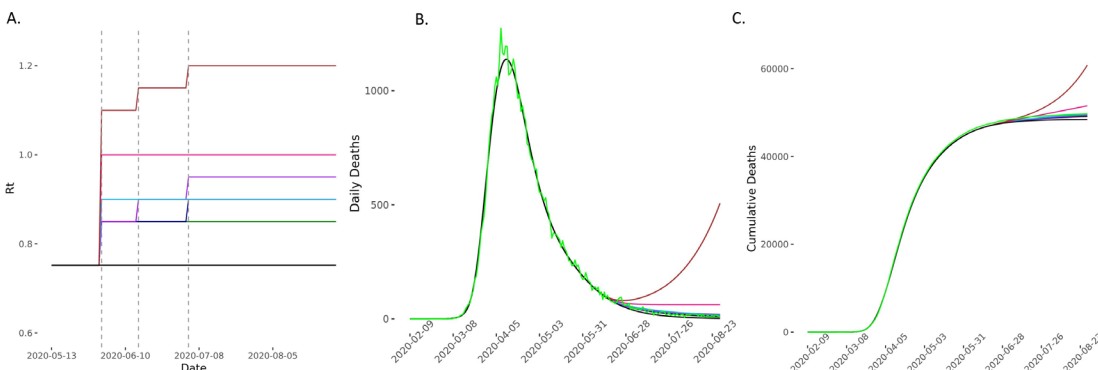

**Figure 9** Predicted deaths in different scenarios of $R_t$ increase on 1 June, 15 June and 4 July compared with the baseline scenario, and real observed death data from the ONS (light green). The model compared scenarios in which $R_t$ increases to different values on 1, 15 and 4 July with real observed deaths (light green). The comparator baseline scenario is of $R_t$ remaining at 0.752 (black). Vertical dashed lines represent time points of easing lockdown. (B,C) The incident and cumulative deaths increase in all scenarios in which $R_t$ increases. The daily deaths appear to fit best with the scenarios where $R_t$s are between 0.85 and 0.95 (dark blue, light blue and purple) during this period. ONS, Office of National Statistics; $R_t$, time-varying reproduction number.

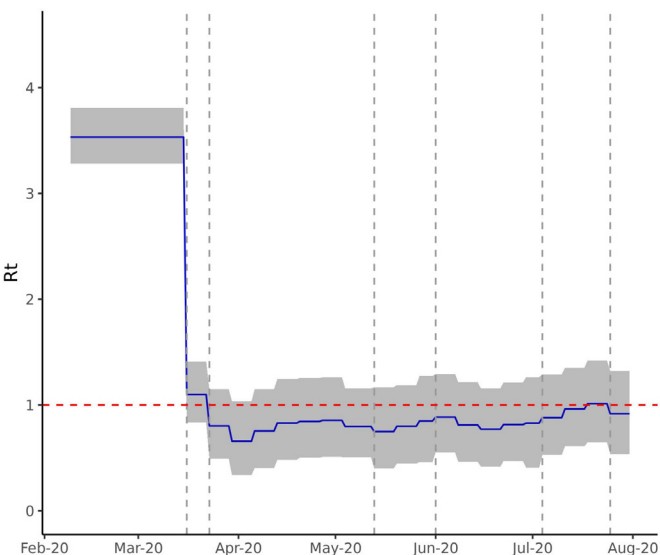

**Figure 10** Estimated time-varying reproduction number ($R_t$) for England. The figure shows the $R_t$ estimated from the recent ONS death data (up to 11 September 2020) with 95% credible intervals (grey) with a serial interval of mean 4.7 and SD 2.9 days. We see a gradual upward trend in inferred $R_t$, with median $R_t$ rising above 1 toward the end of July. ONS, Office of National Statistics.

greater deviations from true estimates would make our forecasting less reliable. Third, similar to Flaxman *et al*,[5] our model uses the IFR as a multiplier for the distribution of time from infection to death, in the absence of reliable population-level case fatality rates (CFRs). While this would not affect the estimation of deaths, if the CFRs were higher (due to large proportions of cases being asymptomatic), then the predicted case numbers would be overestimated by our model. We note, however,

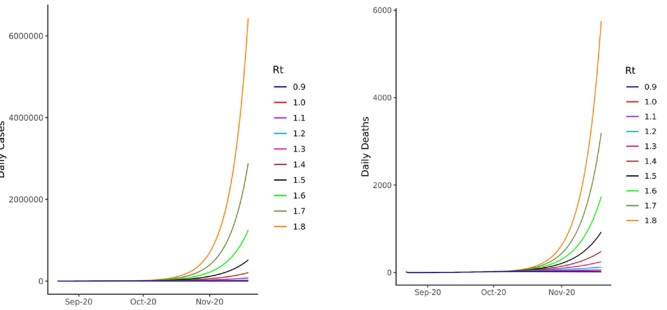

**Figure 11** Predicted cases and deaths at different $R_t$ values from current case numbers in England as of 21 September 2020. Figure 11 represents the predicted rise in cases based on different $R_t$ values, and a serial interval of mean 4.7 and SD 2.9 days. The case numbers were calculated as a moving 7-day average from the Public Health England data of confirmed cases within England up to 21 September. We project case and death numbers (assuming an IFR of 1%) from these incident case numbers, using different scenarios of $R_t$. We note that case numbers are likely underestimates, as the testing system within England is currently running at capacity, and not everyone with symptoms is able to access tests. IFR, infection fatality rate; $R_t$, time-varying reproduction number.

that the estimate of IFR we used (1.1%) is consistent with the CFR estimated previously from Beijing[16] and Flaxman *et al*.[5] We have also, for simplicity, assumed that IFR remains constant throughout the pandemic and the forecasting period. Given that age is an important determinant of mortality, our model may not reflect the changes in the age composition of infected individuals, and changes in healthcare and treatments over time, influencing the accuracy of inference and forecasting. Unfortunately, the ONS does not provide age-stratified daily death data for England to allow us to model differences in age structure. We have, therefore, not considered these in our inference or forecasting. We note that if cases occur disproportionately in younger populations following easing of lockdown, excess deaths may be overestimated during our forecasting period. Fourth, we did not consider the impact of mitigatory measures in our current modelling. However, as we have seen, mitigatory measures were implemented with significant delays from when community transmission increased, as many experts had expected. Nevertheless, if implemented with sufficient rigour and coverage, mitigatory measures would reduce the impact of the modelled scenarios. We note that our inferred $R_t$ based on recent death data should reflect the impact of mitigatory measures, such as testing, contract tracing and isolating, as well as mask use, as inferred $R_t$ values were allowed to change every week. Finally, we only modelled a limited set of scenarios, mainly restricted to those in which $R_t$ remained ≤1.2, but there are multiple possible scenarios that could be modelled. We note that the scenarios modelled are in line with $R_t$ ranges that were subsequently inferred from current death data.

In summary, we show that increases in $R_t$ as a result of easing lockdown would have a substantial impact on incident transmission and deaths for even modest increases that still maintain $R_t$ ≤1, and an even greater impact should $R_t$ rise above 1. This has subsequently been borne out by the observed data. Our findings and the observed data thus far argue strongly for a much more cautious approach in public health management, an urgent need for a properly functioning test, trace and isolate system, with adequate support for isolation,[17] robust mitigatory measures in schools[18] and serious consideration of elimination strategy alongside vaccine roll-out to control the pandemic. Such a multipronged approach aimed at elimination is necessary and its value has been clearly demonstrated in terms of lower case numbers, fewer deaths and lower economic impacts in countries that have followed such strategies.[19] This is all the more important given that continuing transmission has favoured adaptation of SARS-CoV-2, with emergence of several variants of concern some of which are more transmissible, more able to escape immunity from vaccines or both. Elimination allows us to reduce uncertainty associated with new variants and conserves vaccine effectiveness by preventing emergence of new variants that may threaten this.

**Acknowledgements** We would like to acknowledge and thank Flaxman *et al* and the team at Imperial College London for making their code available for us to use.

**Contributors** DG conceived the study and designed the model with NS. DG programmed the model and made the figures. HZ and NS consulted on the model design. All authors interpreted the results, contributed to writing the article and approved the final version for submission.

**Funding** HZ is partly funded by the Bernard Wolfe Health Neuroscience Fund at the University of Cambridge. NS is funded by a Strategic Award from the Wellcome Trust (208363/Z/17/Z). DG is funded by the UKRI/Rutherford HDR-UK fellowship programme (reference MR/S003711/2), and the NIHR AIM Development award (reference NIHR202646).

**Competing interests** None declared.

**Patient consent for publication** Not required.

**Provenance and peer review** Not commissioned; externally peer reviewed.

**Data availability statement** Data are available in a public, open access repository. All data on daily deaths used in this study were taken from the Office of National Statistics website (https://www.ons.gov.uk/peoplepopulationandcommunity/birt hsdeathsandmarriages/deaths/datasets/weeklyprovisionalfiguresondeathsregister edinenglandandwales). The code for the model and dataset analysed are available at: https://github.com/dgurdasani1/lockdownsim.

**ORCID iD**
Deepti Gurdasani http://orcid.org/0000-0001-9996-6929

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
