## [Reviewer comments · BMJ Open]

ARTICLE DETAILS

TITLE (PROVISIONAL)	Modelling the impact of lockdown easing measures on cumulative COVID-19 cases and deaths in England
AUTHORS	Ziauddeen, Hisham; Subramaniam, Naresh; Gurdasani, Deepti

VERSION 1 – REVIEW

REVIEWER	Peixoto, VASCO RICOCA FREIRE DUARTE Universidade Nova de Lisboa, Nova National School of Public Health
REVIEW RETURNED	04-Aug-2020

GENERAL COMMENTS	Relevant and well justified with adequate discussion of limitations. Consider referring in the discussion : COVID-19 severe outcomes disproportionately affect the elderly. Modell assume a certain distribution of cases by age group(equal to that of previous cases? because the model does not consider it)However, if strategies to protect the most vulnerable to outcome "death" are effective, this scenarios may be overestimated . differential impacts of easing lockdown in different age groups must be refered and considered. Age is not refered as the greatest determinant of death by covid-19 it is natural that while easing restrictions more cases will happend among younger population with much smaller risk of death. I would add a paragraph in the end of the discussion with implications of forecasts in terms of health care demand Hospital ICU in face of England's capacity.
--

REVIEWER	Lonergan, Mike University of Dundee
REVIEW RETURNED	12-Aug-2020

GENERAL COMMENTS	The authors fit a Bayesian model to numbers of reported deaths in England up to 12th June, then use it to predict numbers of deaths over the next 90 days, given different possible values of R. As far as I can tell, the predictions are mainly connected to the model through the state of the population on the 12th June. The predicted values are essentially the product of multiplying the expectation of the number dying on 12th June by a truncated power series of R ($R+R^2+R^3+\dots R^n$) representing subsequent generations of infection.
--

The assumption that R remains below 1 after the easing of lockdown may limit the utility of the predictions.

An advantage of the Bayesian model is the possibility it gives to represent uncertainty, but comparing supplementary figure 3 to figure 1 shows that the priors on R are important to this. The priors were based on "estimates of time changing R_t from the University of Cambridge MRC biostatistics nowcasting and forecasting models¹⁰", which I believe also use English mortality data in some form. If that is so, the main results may use some of the data twice and be overly precise.

Figure 2 and supplementary figure 5 are remarkably similar, but have around a 25% difference in R values resulting from the different serial intervals used. I'd expect those differences to produce substantial differences in the predicted numbers of deaths. Do they? If so, the model probably needs to incorporate the uncertainty from the serial interval parameters in the uncertainty in its predictions. The estimate of 3.65 for R_0 is in the tail of the prior (normal mean 2.5, sd 0.5), the 4.25 for the model using the longer serial interval is even further out. How much does the prior pull these estimates down?

The model selection criterion combines two measures: when was that decided, and was there a plan for if they disagreed? Some information on how large a difference in score matters would be useful.

Supplementary table 3 seems to say the models give no evidence that R_t changed at lockdown on 23rd march, but detect evidence that R_t changed on each of the 16th March, the 13th May, and the 1st June. Since the main lockdown announcement was the 23rd March, that seems quite surprising and worth further consideration.

The spike in R_{hat} at about 1.0015 in supplementary figure 1 is striking. I assume it is an artifact, maybe of discretising of distributions? Or does it come from a single chain that stuck? I was also slightly surprised to see "tropical medicine" in the keywords.

Overall, I feel this to be an interesting, but not particularly novel, model of these data. I suspect its results are overly precise, and am not convinced the model is necessary to the estimation of the results presented. There would seem to be easier ways to make those predictions, and probably other questions for which it might be more appropriate.

Response to reviewers' comments:

Reviewer 1:

Reviewer Name: Vasco Ricoca Peixoto

Institution and Country: Nova National School of Public Health, Public Health Research Centre

Please state any competing interests or state 'None declared': None declared

Relevant and well justified with adequate discussion of limitations.

Consider referring in the discussion : COVID-19 severe outcomes disproportionately affect the elderly. Modell assume a certain distribution of cases by age group (equal to that of previous cases? because the model does not consider it) However, if strategies to protect the most vulnerable to outcome "death" are effective, this scenarios may be overestimated. differential impacts of easing lockdown in different age groups must be referred and considered. Age is not referred as the greatest determinant of death by covid-19 it is natural that while easing restrictions more cases will happened among younger population with much smaller risk of death.

We thank Dr Peixoto for raising this important point. We had mentioned this as a limitation of the model in the discussion. Our model assumes constant infection fatality rate, implicitly assuming that age distribution of deaths remains the same over the modelling , and forecasting time period, which may not be the case. In response to this important point raised by Dr Peixoto we investigated the possibility of modeling deaths by age group. Unfortunately, the Office of National statistics does not provide daily age-stratified death data for England so it was not possible to do this. We agree that if the age-distribution of the pandemic during easing lockdown deviates significantly from the distribution currently modelled, we may overestimate deaths, if young individuals proportionately constitute a larger number of cases than modelled. As Dr. Peixoto rightly points out, it may lead to overestimation of deaths in the forecasting period.

We highlight this in our main strengths and limitations section, as follows:

"We have assumed a constant infection fatality rate across time, which would not account for changes in the age-composition of the infected cases over time."

We have also expanded on this point and its implications for our predictions in the discussion on page 12 as follows:

'We have also, for simplicity, assumed that IFR remains constant throughout the pandemic and the forecasting period. Given that age is an important determinant of mortality, our model may not reflect the

changes in the age-composition of infected individuals, and changes in healthcare, and treatments over time, influencing the accuracy of inference, and forecasting. Unfortunately, the ONS does not provide age-stratified daily death data for England to allow us to model differences in age-structure. We have therefore, not considered these in our inference or forecasting. We note that if cases occur disproportionately in younger populations following easing of lockdown, excess deaths may be overestimated during our forecasting period.'

I would add a paragraph in the end of the discussion with implications of forecasts in terms of health care demand Hospital ICU in face of England's capacity.

We thank Dr Peixoto for this suggestion. Given that the original forecasting period has now passed, we have modelled the potential impacts of recent increases in transmission following easing of lockdown and school re-opening on cases and deaths over the next two months, based on the best available estimates of case numbers in England. This is now included in the discussion with a new figure and we have incorporated the point about ICU demand and capacity in this section on page 12:

"In September 2020, the UK is at point where community transmission is once again high and it is clear that we have entered the second wave of the pandemic. Schools reopened in the second week of September, a move that is vitally important to children's health and development, but one that can potentially increase community transmission. Cases and hospitalisations have been increasing exponentially, which has recently translated into an increase in weekly deaths. Using the best available confirmed COVID-19 case data in England published by the UK government on the 21st September (which is likely an underestimate), we modelled the potential impact of increases in transmission on daily cases and deaths over the next two months, assessing different scenarios of increase in R_t . As R_t reaches 1.5, the daily deaths approach 1,000 by late November (Figure 11). We note that the number of deaths forecast during this period could be overestimated if transmission is disproportionately higher among younger age groups, as overall IFR would be lower than the assumed 1%. However, as current data suggests, transmission is likely to spill over into more vulnerable, and older age groups over time. This has profound implications for the health service and the limited ICU capacity available in the NHS, which is at great risk of being overwhelmed. Our modelling suggests that small changes in R_t moving forward could have substantially large effects on case numbers, and deaths, suggesting that mitigatory strategies implemented in a timely manner could have a large impact.'

Reviewer: 2

Reviewer Name: Mike Lonergan

Institution and Country: University of Dundee, UK.

Please state any competing interests or state 'None declared': None declared.

The authors fit a Bayesian model to numbers of reported deaths in England up to 12th June, then use it to predict numbers of deaths over the next 90 days, given different possible values of R. As far as I can tell, the predictions are mainly connected to the model through the state of the population on the 12th June. The predicted values are essentially the product of multiplying the expectation of the number dying on 12th June by a truncated power series of R ($R+R^2+R^3+\dots+R^n$) representing subsequent generations of infection.

We would like to clarify that our model predictions are not the product of multiplying the expected deaths on 12th June by a truncated power series of R. We expand on this below:

Our model is based on the work of Flaxman et al Nature 2020.¹ We used the ONS death data available up to the 12th June to infer the number of initial cases in the pandemic, and changing reproduction number over time till the 29th of May 2020. We then made predictions from the 1st of June, as inferred cases and R_t values after this are unlikely to be reliable (given the time from symptom onset to death distribution).

For each scenario modelled, predicted cases are calculated by multiplying the number of cases at each point by the reproduction number, and the serial interval, which is the distribution of time from infection to transmission, as discussed by Flaxman et al. Therefore, cases are a function of R_t , incident cases on previous days and the SI discretised distribution:

$$c_t = R_t \sum_{j=0}^{t-1} c_j g_{t-j}$$

,

Where t is the forecast time, g_{t-j} is the discretised serial interval distribution probability at $t-j$ days.

Deaths are then calculated as a function of cases on previous days, and the symptom onset to death distribution, as follows:

$$d_t = \sum_{j=0}^{t-1} c_j \omega_{t-j}$$

We have detailed this in the Methods section of the manuscript.

The assumption that R remains below 1 after the easing of lockdown may limit the utility of the predictions.

We thank Dr Lonergan for raising this important point. The primary aim of our work was to examine the impact of easing lockdown in England, even if R was maintained at or below 1 in line with the government's stated aim, to evaluate the impact of a suppression strategy with an elimination strategy. We showed that even with relatively conservative scenarios of increases in R , we would see substantial excess deaths, and cases, suggesting that even an aim of maintaining R at 1, at such high levels of transmission is inadequate. However, we did examine scenarios where R rises up to 1.2 as a result of easing lockdown (Supplementary Tables 3 and 4). We note that R values inferred by SAGE, the MRC forecasting, and nowcasting, and our own inferences now discussed in the revised manuscript suggest the inferred R_t , daily and cumulative death data are within the range of predicted scenarios during the modelled period, and that R rose above 1.1 only toward the end of the forecasting period (August 29th).

We note, however, that R_t values have likely risen above these values recently (in September), leading to the second wave of the pandemic. In this context, we also include forecasts of cases, and deaths from the 21st September over a 60 day period, for a range of R_t values up to 1.8, in line with current estimates of R_t at ~1.4-1.7. We show that small changes in R_t can have substantial impact on the number of cases, and deaths, suggesting that mitigatory strategies at this point could have potentially large impact.

We have now added inferences from modelling R_t values above 1 to the main results, as follows:

“Increases of R_t above 1 at any point resulted in rapid increases in cases, and deaths, with between 3,600-13,000 excess deaths in different scenarios for R_t rising up to between 1 and 1.2, predicting a second wave of the epidemic within England (Figure 4-5 and Supplementary Table 3)”

“For scenarios where R_t rose beyond 1 (up to 1.2), we would expect between 540,000 to 2.8 million excess cases, in line with a second wave (Supplementary Table 3).”

We also discuss the impact of R rising above 1 in the discussion:

“Our model predicted a substantial excess of cases and deaths in several scenarios where R remained ≤ 1 , as well as scenarios where R increased up to 1.2.”

We also discuss prospective forecasts of current cases assuming R_t values up to 1.8.

“In September 2020, the UK is at point where community transmission is once again high and it is clear that we have entered the second wave of the pandemic. Schools reopened in the second week of September, a move that is vitally important to children’s health and development, but one that can potentially increase community transmission. Cases and hospitalisations have been increasing exponentially, which has recently translated into an increase in weekly deaths. Using the best available confirmed COVID-19 case data in England published by the UK government on the 21st September (which is likely an underestimate), we modelled the potential impact of increases in transmission on daily cases and deaths over the next two months, assessing different scenarios of increase in R_t . As R_t reaches 1.5, the daily deaths approach 1,000 by late November (Figure 11). We note that the number of deaths forecast during this period could be overestimated if transmission is disproportionately higher among younger age groups, as overall IFR would be lower than the assumed 1%. However, as current data suggests, transmission is likely to spill over into more vulnerable, and older age groups over time. This has profound implications for the health service and the limited ICU capacity available in the NHS, which is at great risk of being overwhelmed. Our modelling suggests that small changes in R_t moving forward could have substantially large effects on case numbers, and deaths, suggesting that mitigatory strategies implemented in a timely manner could have a large impact.”

An advantage of the Bayesian model is the possibility it gives to represent uncertainty, but comparing supplementary figure 3 to figure 1 shows that the priors on R are important to this. The priors were based on "estimates of time changing R_t from the University of Cambridge MRC biostatistics nowcasting and forecasting models¹⁰", which I believe also use English mortality data in some form. If that is so, the main results may use some of the data twice and be overly precise.

We based our model priors on those used by Flaxman et al, Nature, 2020,¹ which are consistent with those estimated through the Cambridge MRC Nowcasting and Forecasting models, and SAGE estimates. We assessed the sensitivity of the model to priors, and showed that using uninformative priors (no prior

specified) does not substantially impact the inferred values of R_t , although, predictably the uncertainty around the estimates is expectedly increased (Figures 1 and Supplementary Figure 3). We note particularly that R_t values during all time intervals, and particularly so between May-Jun (the period most pertinent to forecasting) are very stable, suggesting that these priors are likely to have minimal effect on our forecasting. This suggests that the priors do not determine the inferred R_t values and therefore are unlikely to affect our inferences from forecasting. We have now included the analysis with uninformative priors in the manuscript, and include this in the results on page 10:

'Using uninformative (no prior specified) priors for R_t did not materially alter the median estimates of R_t , although uncertainty around estimates was predictably increased (Supplementary Figure 3). This suggests our estimates are robust to the priors specified.'

Figure 2 and supplementary figure 5 are remarkably similar, but have around a 25% difference in R values resulting from the different serial intervals used. I'd expect those differences to produce substantial differences in the predicted numbers of deaths. Do they? If so, the model probably needs to incorporate the uncertainty from the serial interval parameters in the uncertainty in its predictions. The estimate of 3.65 for R_0 is in the tail of the prior (normal mean 2.5, sd 0.5), the 4.25 for the model using the longer serial interval is even further out. How much does the prior pull these estimates down?

We thank Dr. Lonergan for raising this point. Specifying longer serial intervals predictably leads to different estimates of excess cases and deaths, which are different to the forecasts from the primary model, with excess deaths predicted in the longer serial interval scenario being higher when compared to the baseline scenario in all scenarios where $R_t < 1.1$ (Supplementary Tables 3 and 4). Our findings are in line with previous work from Flaxman et al (see Supplementary Discuss 3),¹ that also suggested that longer serial intervals can affect inferences of R_0 . As Flaxman et al. note,¹ this difference is likely to be because to reach the currently observed size of the epidemics, a longer assumed generation is compensated by a higher estimated R_0 . We note that while R_0 inferred using the serial interval is higher than that inferred in our primary model with shorter intervals, the R_t values inferred during the lockdown period, and easing of restrictions are remarkably consistent (Figure 1 and Supplementary Figure 4), when inferred with the longer serial interval model. The R_t inferred on the 13th May, and used as the baseline for forecasting is 0.69 with the longer serial interval model, compared with 0.75 in our primary model, a difference of only 8%.

In order to incorporate uncertainty of serial estimates into our estimates, we carried out sensitivity analyses to examine the extent of impact of altering serial intervals on inferred R_t values (Supplementary Figure 4), and forecast excess cases and deaths (Supplementary Table 3 and 4). We discuss this further in the Results and Discussion sections, as follows:

“Using a longer SI leads to an increase in the estimated R_0 , although subsequent estimates following easing of lockdown remain broadly comparable (Supplementary Figure 4). This model is comparable to the primary model with regard to fit to observed deaths (Supplementary Figure 5) although we note that predicted excess deaths and cases in all scenarios where $R_t < 1.1$, are higher than in the primary model with shorter serial interval (Supplementary Table 4), suggesting the primary model is likely to be conservative.”

The model selection criterion combines two measures: when was that decided, and was there a plan for if they disagreed? Some information on how large a difference in score matters would be useful.

The model selection criteria measures were decided a priori. We prioritised ELPD as the estimate for comparison, based on previous work by Dehning et al., in a similar context.² While we also report RMSE, we did not expect large inconsistencies between these. Large inconsistencies would have suggested serious violations in assumptions of ELPD, which would have required further assessment, as to the cause of this. We assessed differences between models based on the SE of the difference, as estimated by ELPD, prioritising simplicity in models that were not significantly different (i.e. models that required fewer parameters to be estimated). Based on this, we chose Model 3 as our primary model. We also note that inferences and predictions from Model 2 and Model 3 are very consistent, and that using one model over the other would be very unlikely to alter our estimates or inferences. We expand on this in the Methods section of the manuscript, as follows:

“For each model, we used the R package loo to calculate expected log pointwise predictive density (ELPD) using Leave-one-out cross-validation (LOO) individually for each left out data point based on the model fit to the other data points. We then calculated between-model differences in ELPDs, to assess whether particular models predicted data better than others, as discussed previously.³ As the assumptions in estimation of ELPD may be violated given these are time-series data, and therefore correlated, we also compared the root mean squared errors (RMSE) across models to assess fit. The final model used was arrived upon based on these comparisons, prioritising differences in ELPD, as this has been used in a similar context to assess change points, previously.² We assessed whether models were significantly different (ELPD difference/SE of difference > 2). When models were not statistically significantly different in performance, for simplicity, we prioritised the model where the least number of parameters needed estimation.”

Supplementary table 3 seems to say the models give no evidence that R_t changed at lockdown on 23rd March, but detect evidence that R_t changed on each of the 16th March, the 13th May, and the 1st June. Since the main lockdown announcement was the 23rd March, that seems quite surprising and worth further consideration.

Our inferences suggest that a substantial reduction in R occurred after social distancing measures were introduced on the 16th March, and then declined further with lockdown. We note that while there was a decline in R_t after the 16th of March, the estimated R_t remained above 1 at this point, with further decline following lockdown, during which R_t reached a low of 0.66 (95% CI 0.34-1.04) during the week 30th March-5th April 2020. This is consistent with studies that have examined the effectiveness of social distancing in the context of SARS-CoV-2, suggesting that this can have a substantial impact on the growth rate of the pandemic, even without lockdown.⁴ This may also reflect changes in behaviour across the community even prior to lockdown, as have been widely reported, including drops in use Apple Maps for public transport by 58% on the 16th of March.[Link]. We note that the closure of schools on the 20th March may also have contributed to this reduction in transmission prior to lockdown. This is also likely to reflect the complex impact of public behaviour and lockdown on R_t , with different measures (e.g. closure of non-essential businesses, working from home, school closures) having had a potentially more delayed impact, as measures such as working from home may have been implemented more gradually in some parts of the workforce. This is one of the reasons we allowed the R_t to vary every week during lockdown.

The spike in R_{hat} at about 1.0015 in supplementary figure 1 is striking. I assume it is an artifact, maybe of discretising of distributions? Or does it come from a single chain that stuck?

We thank Dr Lonergan for raising this point. The R_{hat} is calculated for all predicted variables, including forecast daily cases, R_t values and deaths. This means that many values are duplicated (e.g. R_t estimates for each period of inference would be the same, but are repeated as duplicates for each day, generating a skewed distribution). We have now removed these duplicates. After removal of duplicated values (considering only independent values of R_{hat} for each inferred parameter estimate), we find the distribution has a mean of 1.000057, consistent with convergence of chains. We now include the new figure in the Supplementary Materials (Supplementary Figure 1), and clarify this in the legend.

I was also slightly surprised to see "tropical medicine" in the keywords.

We have removed 'tropical medicine' from the key words.

Overall, I feel this to be an interesting, but not particularly novel, model of these data. I suspect its results are overly precise, and am not convinced the model is necessary to the estimation of the results presented. There would seem to be easier ways to make those predictions, and probably other questions for which it might be more appropriate.

All models that have carried out forecasting within the UK, including the MRC Nowcasting and Forecasting model, SAGE models, and the Imperial models (Flaxman et al.¹) use death data with either Bayesian mechanistic models, or SEIR based Bayesian models for inference, and forecasting. Our Bayesian mechanistic model of the infection cycle to observed deaths infers plausible upper and lower bounds (Bayesian credible intervals) of the initial case numbers, and the reproduction number over time (R_t), as Flaxman et al. from Imperial have done previously.^{1,5} These models use death data, as case data has been unreliable throughout the UK pandemic due to changes in testing policy, and practice over time. As Flaxman et al. state in their report,⁵ this approach has been used in numerous previous studies⁶⁻⁹ and has a strong theoretical basis in stochastic individual-based counting processes such as Hawkes process and the Bellman-Harris process.^{10,11} The renewal model is related to the Susceptible-Infected-Recovered model, except the renewal is not expressed in differential form.

We hope that we have now clarified how the model is necessary for the estimation of the results and Dr Loneragan is reassured by the comparisons to the observed data, and the robustness of the model to priors.

References

1. Flaxman S, Mishra S, Gandy A, et al. Estimating the effects of non-pharmaceutical interventions on COVID-19 in Europe. *Nature* 2020.
2. Dehning J, Zierenberg J, Spitzner FP, et al. Inferring change points in the spread of COVID-19 reveals the effectiveness of interventions. *Science* 2020.
3. Vehtari A, Gelman, A., and Gabry, J. Practical Bayesian model evaluation using leave-one-out cross-validation and WAIC. *Statistics and Computing*. *Statistics and Computing* 2017; 27(5): 1413-32.
4. Hsiang S, Allen D, Annan-Phan S, et al. The effect of large-scale anti-contagion policies on the COVID-19 pandemic. *Nature* 2020; 584(7820): 262-7.
5. Flaxman S MS, Gandy, A, Unwin HJT, Coupland H, Mellan, TA, Zhu H, Berah T, Eaton JW Report 13: Estimating the number of infections and the impact of non-pharmaceutical interventions on COVID-19 in 11 European countries: Imperial College, London, 2020.
6. Fraser C. Estimating individual and household reproduction numbers in an emerging epidemic. *PLoS One* 2007; 2(8): e758.
7. Cori A, Ferguson NM, Fraser C, Cauchemez S. A new framework and software to estimate time-varying reproduction numbers during epidemics. *Am J Epidemiol* 2013; 178(9): 1505-12.
8. Nouvellet P, Cori A, Garske T, et al. A simple approach to measure transmissibility and forecast incidence. *Epidemics* 2018; 22: 29-35.

9. Cauchemez S, Valleron AJ, Boelle PY, Flahault A, Ferguson NM. Estimating the impact of school closure on influenza transmission from Sentinel data. Nature 2008; 452(7188): 750-4.

10. Bellman RH, T. On Age-Dependent Binary Branching Processes. Ann Math 1952; 55: 280-95.

11. Bellman RH, T. E. On the Theory of Age-Dependent Stochastic Branching Processes. Proc Natl Acad Sci 1948; 34: 601-4.

VERSION 2 – REVIEW

REVIEWER	Peixoto, VASCO RICOCA FREIRE DUARTE Universidade Nova de Lisboa, Nova National School of Public Health
REVIEW RETURNED	16-Oct-2020

GENERAL COMMENTS	The final sentencd of the conclusion could be reconsidered. An elimination strategy is probably unachievable without major lockdowns and they would still be insuficcient in democracies with constitutional protection of individual freedoms. An heavy lockdown strategy would have major economic , social and political impacts that can not be neglected. From recent evidence it is likely that we can sustain a relativley high level of transmission among younger people without major impacts on health services and hospitals. Further than that a compliant population, wearing masks, good test seeking and isolation when symptomatic, social distancing and prohibition of potential superspreading circumstances can maintain transmission in check while avoiding negative impacts of very stringent restrictive measures. This could be explored in the discussion.~ with relevant references. The final message should not be reccomending and elimination strategy without discussing its possible harms. ~ITs should state , as it does, what impact different variations of $R(t)$ could have and use it to reinforce the need to compensate the easing of lockdown measures with more and better public communication for prevention , improving indivudal preventive behaviour, and risk management, compliance with testing ,contact tracing etc, and investment in capacity to respond to such increases. With a compliant population, with very high case ascertainment, good contact tracing and a set of preventive measures in place and protecting the vulnerable there are examples that it is possible to mantain levels of transmission. Every intervention in health and public health has costs and benefits.
--

REVIEWER	Kelly, Brian Bradford Institute for Health Research, Born in Bradford
REVIEW RETURNED	15-Feb-2021

GENERAL COMMENTS	This is a useful and important piece of analysis. The paper is well presented, clear and well written. The statistics, though complex, are also presented clearly, and fit well with the objective. The additions since the first draft greatly improve the paper; particularly the comparison of the observed data over the original forecasting period and the model predictions. The set of predictions on the potential impact of currently observed exponential rises in cases over the next two months is useful. Also the sensitivity analysis is welcome and well presented. The COVID pandemic is fast moving, and events quickly overtake research, but this paper has wider implications for pandemic management and contributes to the importance of understanding and learning from the pandemic for the future. I would recommend this paper for publication in it's current form.
---

VERSION 2 – AUTHOR RESPONSE

Reviewer: 1

Dr. VASCO RICOCA FREIRE DUARTE Peixoto, Universidade Nova de Lisboa

Comments to the Author:

The final sentence of the conclusion could be reconsidered.

An elimination strategy is probably unachievable without major lockdowns and they would still be insufficient in democracies with constitutional protection of individual freedoms. An heavy lockdown strategy would have major economic , social and political impacts that can not be neglected.

We thank the reviewer for these comments. We respectfully disagree with the reviewer that elimination strategies are more detrimental with respect to economy and society relative to other strategies. While all strategies for containing SARS-CoV-2 have some social and economic impact, evidence over the past year suggests strongly that social and economic costs of elimination strategies are far lower than those of mitigation strategies. There is a false dichotomy that has been created between public health and economy. However, evidence shows that countries that aimed for elimination had the least impact on their economy, spent the least amount of time in restrictions and lockdowns, and had the fewest number of deaths per population. Deaths in OECD countries that opted for elimination have been 25 times lower than those that favoured mitigation. Among OECD countries, liberties were most severely impacted in those that chose mitigation, whereas swift lockdown measures—in line with elimination—were less strict and of shorter duration.¹

Elimination has been achieved in many different contexts and countries, including Vietnam, Thailand, Taiwan, China, Iceland, Australia and New Zealand. Currently, 30 countries are elimination zones (<https://www.endcoronavirus.org/countries#winning>). Even countries where elimination was aimed for and not achieved have generally performed better with regards to public health, economy, and time spent in lockdown and restrictions.¹ Evidence suggests that there are benefits of aiming for elimination,

rather than mitigation, even if this is not achieved. We are not aware of any harms of the elimination strategy, as it not only prevents deaths, long COVID, but also reduces time spent in restrictions and economic impacts.

Figure COVID-19 deaths, GDP growth, and strictness of lockdown measures for OECD countries choosing SARS-CoV-2 elimination versus mitigation

Reproduced from the Lancet.¹

It is also not true that elimination strategies would not be possible in democracies with constitutional protection of individual freedoms.¹ They require less time spent in restrictions compared with mitigation strategies, and have been achieved in democracies where individual freedoms are prioritised, such as New Zealand, Australia and Iceland. Indeed, data from the UK has consistently shown high support for lockdowns at every point, including earlier this year, and opposition to early easing of lockdowns.² Despite widespread coverage of instances of lack of compliance in the press, adherence to stringent behavioural regulations has remained extremely high (over 90%), even though many people are suffering considerably, both financially and psychologically.² Where compliance has been low, e.g. with isolation, this has been related to difficulty in isolating due to lack of financial support rather than lack of motivation or understanding the need to do this. Current research suggests that only 18-22% of those who have symptoms seek testing because of concerns about not being able to isolate due to poor financial support.³ The UK currently has the lowest sick pay across all OECD countries, and it is key that more support is provided to allow those who are most exposed to seek tests when ill, and be able to isolate without financial repercussions.

From recent evidence it is likely that we can sustain a relatively high level of transmission among younger people without major impacts on health services and hospitals.

While younger age groups are less likely to be hospitalised or die from long COVID, it is clear that high levels of transmission in these groups can have important impacts on health. Recent data released from the Office for National Statistics suggest there are an estimated 1.1 million people in the UK living with long COVID, most of whom are young, and did not have pre-existing conditions. Two-thirds of these reported impact on day-to-day activities and almost half a million have had symptoms lasting 6 months or more. 20% of young people go on to develop long term symptoms lasting 5 weeks or more following infection, the consequences of which are unknown.⁴ Evidence suggests that infection, even among younger people may be associated with long-term organ dysfunction, and immune dysregulation. Given these impacts it would be unwise to expose younger people to high levels of infection, and consider deaths as the only poor outcome from COVID-19.

Aiming to keep transmission just low enough to prevent hospitals being full has had huge impacts on public health, with over 150,000 deaths within the UK. Despite NHS capacity not being reached in the two waves to date, defined as those needing urgent care being able to access this, routine care has been substantially affected with millions of people now awaiting routine care as a result of this approach.⁵

Another consequence of allowing high levels of transmission is that this can lead to virus adaptation. Several new variants of concerns have emerged in regions of high transmission over the past year, many of which have been more transmissible, and more fatal, and some that have been associated with lower effectiveness of vaccines at least against mild to moderate disease.⁶ The recent Kent variant has contributed to the current third wave occurring in Europe. The P.1 variant has been linked to the recent surge of infections in Brazil which has overwhelmed hospitals and led to thousands of deaths. More recently the delta variant (B.1.617.2) has spread rapidly across the UK, with evidence suggesting it is 50-70% more transmissible,⁷ with some level of escape from vaccines. This is now resulting in early exponential growth in cases, accompanied by increases in hospitalisations within the UK, with predictions from SAGE suggesting that this could lead to a level of hospitalisations similar or exceeding the January 2021 wave in the UK.

Modelling also suggests that allowing transmission to continue at high levels while rolling out vaccination predominantly among vulnerable populations would provide favourable conditions for virus adaptation towards escape from vaccines.⁸ In this context, controlling transmission is even more important to prevent new variants emerging, and also to protect our precious vaccine resources.

Further than that a compliant population, wearing masks, good test seeking and isolation when symptomatic, social distancing and prohibition of potential superspreading circumstances can maintain transmission in check while avoiding negative impacts of very stringent restrictive measures. This could be explored in the discussion.~ with relevant references.

We agree that a multi-pronged approach is needed. There are several countries that followed elimination strategies early on without stringent lockdowns, by good public health messaging, focus on preventing aerosol transmission, social distancing, widespread mask use and excellent backward and forward contact tracing systems. Using these approaches kept transmission under control, negating the need for national lockdowns.

The final message should not be recommending an elimination strategy without discussing its possible harms. ~ITs should state, as it does, what impact different variations of $R(t)$ could have and use it to reinforce the need to compensate the easing of lockdown measures with more and better public communication for prevention, improving individual preventive behaviour, and risk management, compliance with testing, contact tracing etc, and investment in capacity to respond to such increases. With a compliant population, with very high case ascertainment, good contact tracing and a set of preventive measures in place and protecting the vulnerable there are examples that it is possible to maintain levels of transmission.

Every intervention in health and public health has costs and benefits.

We agree that good public communication, test trace isolate and support systems, and focus on air cleaning are all essential to pandemic control. We agree that protecting the vulnerable, through shielding, and vaccination should be part of any strategy, but is not in itself sufficient and transmission needs to be kept under control across all age groups. Evidence suggests that young healthy individuals can suffer long-term impacts from long COVID, and transmission across younger or less vulnerable groups inevitably spills over to vulnerable groups. Given vulnerable groups constitute up to 30% of the population in many countries, a 'focused protection' strategy would not be possible, as has been discussed before.⁹ Even if there was any doubt about pursuing elimination strategies at the beginning of the pandemic, the evidence from across the globe over the last year makes it abundantly clear that elimination is the most optimal strategy. This is all the more important given the emergence of new SARS-CoV-2 variants which threaten effectiveness of pandemic response, and vaccine effectiveness.

Elimination would prevent emergence and import of new variants, greatly reducing the risks posed by these during vaccine roll-out. We therefore respectfully disagree with the reviewer on this point.

We have amended the final paragraph of the manuscript as follows:

'Our findings and the observed data thus far argue strongly for a much more cautious approach in public health management, an urgent need for a properly functioning test, trace and isolate system, with adequate support for isolation,¹⁰ robust mitigatory measures in schools¹¹ and serious consideration of elimination strategy alongside vaccine-roll out to control the pandemic. Such a multi-pronged approach aimed at elimination is necessary and its value has been clearly demonstrated in terms of lower case numbers, fewer deaths and lower economic impacts in countries that have followed such strategies.¹ This is all the more important given that continuing transmission has favoured adaptation of SARS-CoV-2, with emergence of several variants of concern some of which are more transmissible, more able to escape immunity from vaccines, or both. Elimination allows us to reduce uncertainty associated with new variants, and conserves vaccine effectiveness by preventing emergence of new variants that may threaten this.'

Reviewer: 3

Dr. Brian Kelly, Bradford Institute for Health Research

Comments to the Author:

This is a useful and important piece of analysis.

The paper is well presented, clear and well written. The statistics, though complex, are also presented clearly, and fit well with the objective.

The additions since the first draft greatly improve the paper; particularly the comparison of the observed data over the original forecasting period and the model predictions. The set of predictions on the potential impact of currently observed exponential rises in cases over the next two months is useful. Also the sensitivity analysis is welcome and well presented.

The COVID pandemic is fast moving, and events quickly overtake research, but this paper has wider implications for pandemic management and contributes to the importance of understanding and learning from the pandemic for the future.

I would recommend this paper for publication in its current form.

We thank the reviewer for their helpful and positive comments and are glad that our revisions have addressed their concerns,

References:

1. Olliu-Barton M, Pradelski BSR, Aghion P, Artus P, Kickbusch I, Lazarus JV. SARS-CoV-2 elimination, not mitigation, creates best outcomes for health, the economy, and civil liberties. *The Lancet* 2021.
2. Reicher S, Drury J. Pandemic fatigue? How adherence to covid-19 regulations has been misrepresented and why it matters. *The BMJ Opinion* 2021.

3. Smith LE, Potts HWW, Amlot R, Fear NT, Michie S, Rubin GJ. Adherence to the test, trace, and isolate system in the UK: results from 37 nationally representative surveys. *BMJ* 2021; 372: n608.
4. Statistics OfN. Prevalence of ongoing symptoms following coronavirus (COVID-19) infection in the UK: 1 April 2021.
5. Campbell D MS. Covid: 4.6m people missed out on hospital treatment in England in 2020. *The Guardian*. 2021.
6. Madhi SA, Baillie V, Cutland CL, et al. Efficacy of the ChAdOx1 nCoV-19 Covid-19 Vaccine against the B.1.351 Variant. *N Engl J Med* 2021.
7. England PH. SARS-CoV-2 variants of concern and variants under investigation in England: Technical briefing 13, 2021.
8. Gog JR HE, Danon L, Thompson R. Vaccine escape in a heterogeneous population: insights for SARS-CoV-2 from a simple model. *MedRxiv* 2021.
9. Alwan NA, Burgess RA, Ashworth S, et al. Scientific consensus on the COVID-19 pandemic: we need to act now. *Lancet* 2020; 396(10260): e71-e2.
10. Gurdasani D, Bear L, Bogaert D, et al. The UK needs a sustainable strategy for COVID-19. *Lancet* 2020; 396(10265): 1800-1.
11. Gurdasani D, Alwan NA, Greenhalgh T, et al. School reopening without robust COVID-19 mitigation risks accelerating the pandemic. *Lancet* 2021; 397(10280): 1177-8.